# It’s about Time: Ossification Center Formation in C57BL/6 Mice from E12–E16

**DOI:** 10.3390/jdb6040031

**Published:** 2018-12-15

**Authors:** Kevin Flaherty, Joan T. Richtsmeier

**Affiliations:** 1Department of Anthropology, Pennsylvania State University, University Park, PA 16802, USA; jta10@psu.edu; 2Department of Biology, Augustana College, Rock Island, IL 61201, USA

**Keywords:** ossification center, staging system, heterochrony, morphometrics, OSX, embryogenesis, mouse models, eMOSS

## Abstract

The establishment of precise, high-resolution temporal sequences for morphogenetic events in laboratory mice remains a vexing issue in developmental biology. Mouse embryos collected at the same period of gestation, even those from the same litter, show wide variation in individual levels of progress along their developmental trajectory. Therefore, age at harvest does not provide sufficient information about developmental progress to serve as the basis for forming samples for the study of rapidly or near-simultaneously occurring events such as the sequence of ossification center formation. Here, we generate two measures of individual developmental progress (developmental age) for a large sample of mouse embryos using crown–rump lengths that measures size, and limbstaging ages produced by the embryonic Mouse Ontogenetic Staging System (eMOSS) that measure shape. Using these measures, we establish fine-grained sequences of ossification center appearance for mouse embryos. The two measures of developmental progress generate slightly different sequences of ossification center formation demonstrating that despite their tight correlation throughout the developmental period, size and shape are aspects of form that are at least partially dissociated in development.



**“Time is the longest distance between two places.”**
—Tennessee Williams, The Glass Menagerie


## 1. Introduction

The establishment of precise, high-resolution temporal sequences for morphogenetic events in laboratory mice, currently the most extensively used experimental mammal for studying human development and disease, remains a vexing issue in developmental biology. One of the main obstacles to establishing a precise developmental timeline for any developmental event is the high level of variability in developmental progress exhibited by embryos collected at similar days of gestation. Mouse embryos collected at the same embryonic age, even littermates, show disparities in morphological progress indicating variation in the rate of development [1,2,3]. As a result, for any given developmental process (e.g., palate elevation, heart septation, limb bud development, organogenesis, etc.), each mouse within a litter will exhibit a different amount of progress along the trajectory towards completion of that process. Similar developmental differences appear across litters collected at the same time, with some litters appearing older or younger in aggregate than other litters. Consequently, selecting specimens to analyze molecular or cellular details of rapidly occurring developmental phenomena based solely on age information determined from timed matings is problematic. Inter-individual differences in rates of development make it difficult to select individuals to represent precise stages of development, even when matings are conducted on short-duration timelines and when time-consuming and often subjective examination of developmental morphology is used to stage individuals.

One example of the difficulty in establishing precise temporal sequence is the initial establishment of the mineralized skeleton. Dermal bone is not preformed in cartilage, but forms directly through intramembranous ossification [4,5] that involves the differentiation and proliferation of osteoprogenitor cells directly from mesenchyme. The initiation of skeletogenesis of dermal bones of the cranial vault and facial skeleton (the dermatocranium) is marked by ossification center differentiation, while skeletogenesis of endochondrally forming bone (the chondrocranium and the majority of post cranial skeletal elements) is marked by the initiation of mineralization of cartilage models. Despite nearly a century of research into bone development using laboratory mice as a model, few researchers have precisely characterized the developmental timing and sequence of ossification center appearance in mice. Even the meticulous atlases of mouse development compiled by Theiler [6] and Kaufman [7] do not attempt to establish an absolute sequence of ossification center development in the mouse.

The production of an embryo represents an assembly of diverse developmental events within any specific developmental process whose sequence is obscured due to the rapidity and variability of mouse development. Skeletogenesis is a developmental process marked by a series of developmental events: the proliferation, condensation, and differentiation of mesenchymal cells to a chondrogenic or osteogenic fate [8], continued proliferation of those cells, and the production and mineralization of osteoid by osteoblasts. The appearance of ossification centers for cranial bones can occur rapidly, being separated by less than two hours of embryonic development, which confounds even short duration (commonly two hour) mating schemes (e.g., [9]). Moreover, mice within a litter can display a range of developmental morphologies that exceed what can be achieved during two hours of gestational development. Consequently, since each embryo grows at its own rate and represents a slightly different stage of development, we cannot determine the precise sequence and timing of many developmental events strictly by using timed conceptions [10]. Establishing a reliable read-out of the sequence of ossification center appearance requires a method for sorting embryos according to the relative level of development [2,3].

One potential method for sorting embryos is the use of crown–rump length (CRL), a simple and easy-to-collect metric of the maximum total length of an embryo, to order specimens according to size (Figure 1). Ferguson [11] found that CRL predicted the level of palatal development in rats better than embryo weight and other linear measurements. The precise nature of the relationship between size measured by CRL during embryogenesis and developmental progress is not clear, however. It is possible, despite the high correlation between size and developmental progress, that these two variables are autonomous for some developmental processes, and that CRL measurements cannot be used to generate relative sequences for some aspects of development. Despite the fact that embryos increase in size as their development progresses, the advancement of any individual developmental process (e.g., palate development) may not inherently depend on or be correlated with overall size increase.

Another solution to determining the precise sequence and timing of specific developmental events is the implementation of staging systems that use developmental morphology to establish a metric of the relative level of development of a mouse embryo (developmental age). Developmental age estimates provide information about the stage of development of an individual that cannot be determined from the amount of time elapsed between conception and embryo collection (age at harvest). Staging systems enable researchers to estimate a developmental age for each individual, sort individuals on the basis of developmental ages, and determine the precise sequence of a set of developmental events. There is no single method for determining developmental age for a mouse embryo, as any aspect of embryonic morphology that changes in synchrony with the level of overall development provides a potential metric for measuring developmental age. Thus, if methods are precise, staging systems based on the developmental morphology of the head [1] and those based on the limb [2,12,13] are equally valid for assigning developmental ages to embryos. However, some may be more or less appropriate for particular uses (e.g., head morphology-based staging systems may prove to be better predictors of palatal development relative to limb-based systems).

Though many systems for staging mouse embryos exist [1,6,12,13], they are not routinely used because they are not quantitative and thus contain implicit bias, are low resolution (grouping embryos by embryonic days or half days) providing no better precision than time elapsed since fertilization, or are too time-consuming to be of practical use for most research programs. The embryonic Mouse Ontogenetic Staging System (eMOSS) [3] is a high-resolution staging method that uses the two-dimensional outline of an embryo’s hindlimb to provide an estimate of the relative level of development for embryos collected on embryonic days 10 through 15 (E10–E15). eMOSS age estimates (hereafter called limbstaging ages) are scaled to developmental timing, so that limbstaging ages are provided as point estimates in embryonic days and hours, with an associated confidence interval given as a range of hours of development. eMOSS estimates can be generated rapidly, since the method is scale-invariant and only a picture of the hindlimb is required to establish a limbstaging age estimate. eMOSS is particularly well-suited to the task of establishing a sequence for ossification center appearance, as the process of ossification begins and a large number of ossification centers appear during the embryonic period covered by eMOSS.

The distinction between these two classes of systems for ordering embryos is that one measures the extent of development on the basis of a size metric (CRL) while the other uses a metric designed to represent shape (limb outline) to track development. In evolutionary developmental biology, changes in the timing or rate of developmental events, known as heterochrony [14,15,16,17], lead to changes in size and shape that affect the overall morphology of the organism. Previous investigations into evolution by heterochrony assume that growth and development, represented mathematically by change in size and change in shape, respectively, are evolutionarily dissociable processes [15,16,17]. A large literature rests on this assumption, though little evidence has been gathered on the dissociability of size and shape (as proxies for growth and development) among individuals of the same species.

Here, we use precise data collected from laboratory mice to determine the magnitude of variation exhibited by mouse embryos of the same age at harvest to determine variability in growth and development across our sample. We stage our samples of embryos using information on change in size (using crown–rump length) and change in shape (using eMOSS limbstaging ages) to establish a detailed sequence of cranial ossification center appearance in C57BL/6 and in Osx-GFP mice in which the expression machinery of the Osx (Osterix) gene drives expression of a green fluorescent protein (GFP) reporter gene. We examine the difference in sequences of ossification center appearance in mouse embryos using these two systems. Our use of these two systems enables the establishment of concrete timelines for the appearance of ossification centers using two different aspects of the initiation of ossification centers (osteoblast differentiation and bone mineralization), and provides an indication of the delay between these two events in the development of an ossification center. Our data also provide the first experimental evidence of the dissociability of size and shape in developing mice.

## 2. Results

### 2.1. Analysis of Size Indicates Variation in Growth

Mouse embryos ranging in age from E12 to E16 exhibit considerable variation in overall size. The mean, standard deviation, and range of CRL for each embryonic day (Table 1) are similar to those obtained by Otis and Brent [18], though the ranges of our data are larger, particularly for the oldest specimens. Analysis of variance (ANOVA) showed a significant litter effect across all ages, as well as significant associations of CRL with both harvesting age and limbstaging age (Table 2). The relationships between CRL and both age descriptors is expected, given the inevitable correspondence between the increase in size and increasing complexity of developmental anatomy. More surprising is the significant overlap in CRL ranges between consecutive embryonic days (Figure 2). The range for each day overlaps with the range of the preceding and following day, and average CRL for each day corresponds roughly with the minimum CRL for the following day (e.g., the average for E13 is approximately the minimum for E14). This overlap in size measurements between embryos collected 24 h apart illustrates that the relationship between size and age is highly variable during this period of embryonic development.

### 2.2. Analysis of Shape Indicates Variation in Development

Developmental shape, measured by the limbstaging ages generated by eMOSS, shows substantial variation during this period. This means that mice collected at the same period of gestation exhibit a substantial range of variation in limbstaging scores. Individual embryos show differences between limbstaging age and their expected age based on age at harvest greater than 24 h (Figure 3 and Figure 4). For instance, one E12 specimen had a limbstaging age that was 24.5 h later than the expected age based on age at harvest, indicating that the individual had developed at such an advanced rate relative to a typically developing individual that its hindlimb shape was in the normal range for a mouse collected on E13. Analysis of variance of our shape metric (Table 3) shows a strikingly similar pattern to that of size, with age, size, and litter all showing significant relationships with shape.

### 2.3. Relationship between Size and Shape

There is a generally strong positive relationship between CRL and limbstaging age when all data are considered (R^2^ = 0.92), though the correlation between these variables for each embryonic day is often much lower and exhibits significant daily variation, with R^2^ values ranging between 0.53 and 0.8 (Table 4). It is notable, however, that the spread of variation for CRL and limbstaging age appears to move in opposite directions, with limbstaging age being relatively variable for younger mice in the sample and diminishing on days E14 and E15. CRLs, on the other hand, are more tightly clustered in the younger mice, but show increased variation in the older mice (Figure 5).

### 2.4. Size, Shape, Age, and Bone Development

We collected Osx-GFP mice at six-hour intervals from E12.0–E15.0 to establish high-resolution timelines for the appearance of ossification centers indicated by osteoblast differentiation. C57BL/calcein mice were collected every six hours from E14.0–E16.0 to examine the timeline for the appearance of ossification centers based on bone mineralization. Figure 6 shows that arranging mice based on increasing CRL or limbstaging age produces a significantly different sequence of the appearance of ossification centers relative to the sequence based on age at harvest. This results from the fact that each embryo in a litter is by definition assigned the same age at harvest but may exhibit differences in CRL and limbstaging age. Of the three metrics (age at harvest, CRL, and limbstaging age), limbstaging age appears to provide the most reasonable sequence of ossification center expansion, as mice with less developed hindlimbs (younger limbstaging ages) show smaller ossification centers than those with more developed hindlimbs (older limbstaging ages). When specimens are ordered according to limbstaging age, ossification centers expand in an orderly sequence, with developmentally younger mice having smaller ossification centers and developmentally older mice having larger ones. CRL also provides a logical sequence of ossification center expansion, with smaller mice tending to have smaller ossification centers and larger mice having larger ones. Age at harvest, on the other hand, provides a poor sequence of ossification center expansion, as several mice appear to be developing significantly faster or slower than would be expected based on their age at harvest. In Figure 6, mice collected at E13.25 appear to be significantly younger than those collected at E13.00, as well as some of those collected at E12.75. It is notable that age at harvest appears to be a relatively coarse measure for development in terms of ossification center size compared to either limbstaging age or CRL. Age at harvest is the same for each embryo in a litter but does not reflect the fact that some litters may develop faster or slower than others. Limbstaging age and CRL reveal individual and inter-litter variation and order specimens accordingly. Thus, age at harvest cannot account for individual and inter-litter variation, resulting in the absence of an orderly sequence of ossification center expansion where ossification center size gets progressively larger over time.

Ordering mice based on CRL and based on limbstaging age produce similar, but not identical, ossification center appearance sequences for the Osx-GFP mice (Figure 7) and the C57BL/calcein mice (Figure 8). This suggests that size and shape reflect different aspects of developmental progress, despite their strong correlation. In addition, both CRL and limbstaging age provide higher resolution timelines for bone development than age at harvest. Embryos collected on each embryonic day during this period exhibit wide variation in the rate of development which limbstaging age and CRL can capture but age at harvest cannot.

## 3. Discussion

Establishing temporal priority for a series of developmental events using laboratory mice presents a significant obstacle in developmental research due to variation in the rate of development among individuals. A synchrony of developmental events in mice contributes to the complexity. Unravelling the complex nature of sequential and causally-linked events such as gene expression and cellular differentiation is crucial to understanding the developmental processes that comprise morphogenesis. To understand the initiation of a developmental event, it is often necessary to detect changes in gene expression patterns that may be limited in time and therefore difficult to capture using standard mating schemes. Here, we have shown that measures of growth (CRL) and development (limbstaging age) provide significantly higher resolution timelines for a relatively obvious and binary developmental event (ossification center appearance) than the age at harvest for mouse embryos. Moreover, our data establish a framework for breeding for selected timepoints for research into osteoblast differentiation and mineralization by providing more precise timelines of when specific ossification centers emerge. The methods established here are easily transferable for characterizing the temporal sequence of other events of interest. Limbstaging methodology is readily applicable to establishing temporal priority of gene expression sequences, patterns that shift rapidly during developmental events [19], yet are critical to understanding the nature of the relationship between genotype and phenotype.

Previous staging methodologies have not generally been used to provide the type of variation and developmental sequence data presented here for two main reasons. First, many of the staging systems lack the temporal precision needed to properly sort individual specimens. The Theiler staging system, for instance, describes the major developmental events on each embryonic day, but does not provide sufficient information to sort embryos according to specific levels of development within embryonic days [6]. Second, while other staging systems possess the temporal resolution to correctly sort specimens according to size or developmental morphology, they lack the ease of collecting CRL or eMOSS data. Most prominent among these is the craniofacial development staging system outlined by Miyake et al. [1] that established substages within the Theiler system for E11–E14. Over this range, Miyake et al. found similarly high levels of variation for embryos collected at the same harvesting age. However, scoring the level of development for individual embryos using this system requires high-resolution images of the craniofacial region of embryos as well as the knowledge that enables a researcher to discriminate among subtle differences in craniofacial developmental anatomy. While CRL data has always been available to researchers using inbred mice, it has not been used to establish temporal sequences and variation data outside of a few notable exceptions [11,18].

In a broader context, we have demonstrated that, although size and shape are coupled during development, this relationship is not fixed. Size and shape are highly correlated over this sample whose chronological ages span E12–E15 (R^2^ = 0.92), but the correlation varies across harvesting age groups (ranging from R^2^ = 0.53–0.8). Additionally, we have shown that size and shape show contrasting patterns of variation during this age period for our metrics. Variation in CRL increases with age, while variation in shape (limbstaging age) decreases with age, supporting previous findings of other researchers who studied variation in CRL [18] and in craniofacial development [1].

Finally, the sequence of appearance of ossification centers summarizing data collected from Osx-GFP and C57BL/calcein mice establish slightly different ontogenetic trajectories based on whether the specimens are ordered according to size or shape, though both measures present a more reasonable sequence than that obtained by using harvesting age. This indicates that for research projects seeking to establish temporal priority in developmental events, the metric used to sort individual embryos can seriously impact the results and their interpretation. It is evident that no single “best metric” exists for sorting individuals according to their developmental morphology. Sorting our data according to both limbstaging age (shape) and CRL (size) resulted in highly detailed and reasonable timelines of ossification center appearance, but the timelines are not the same. There may be no unique way to precisely measure the level of development of an individual embryo, and developmental processes may each bear a certain level of dissociation. The detected pattern of dissociability may reflect underlying modular patterns of development that facilitate flexibility that contributes to morphological evolution (see an example in opercle development [20]). In short, each developmental event has its own timeline which may or may not correspond with the pace of other developmental events. Consequently, developmental events used to mark developmental progress of specific processes or of varying anatomical systems may not be appropriate across developmental time or across anatomical systems. Whether an internal (pertaining to the development of a particular system) or external (pertaining to general developmental principles) system is appropriate depends largely upon the research question being posed.

## 4. Methods

Litters of C57BL/6J embryos were generated by timed, overnight matings. The mice were bred twice a week, with males being introduced into the cages housing females at 3:00 p.m. and removed the following morning at 9:00 a.m. Pregnancy was verified by visually confirming the presence of a copulatory plug and weighing breeder females daily. Pregnant females were sacrificed at 9:00 a.m. via CO_2_ exposure followed by cervical dislocation. Embryos were removed from their placental sac, washed with saline, and fixed in 4% PFA for 24 h. The embryos were then transferred to 0.01% Sodium azide solution in PBS for preservation. All mouse procedures employed in this research were approved by the Institutional Animal Care and Use Committee of the Pennsylvania State University (IACUC46558, IBC46590).

To test the dissociability of size and shape, a total of 356 C57BL/6J embryos were collected at 24-h intervals for embryonic days 12–16. Ten litters were bred for each embryonic day, with sample sizes for each day ranging between 65 and 86 individual embryos (Table 5).

To establish temporal sequence for ossification center development, two lines were used. Litters of Osx-GFP mice [21] were collected at six-hour intervals from E12.0 through E15.0 to study the appearance of osteoblasts aggregated in developing ossification centers. Osx marks the differentiation of cells into osteoblasts [22]. To study the appearance of mineralized ossification centers, we used C57BL6/J embryos whose mothers were injected with (20 mg/kg of body weight) calcein one hour prior to sacrifice. Litters of C57BL/calcein mice were collected at 6-hour intervals from E14.0 through E16.0 from the injected C57BL6/J dams. Mice from these two lines will be referred to as Osx-GFP mice and C57BL/calcein mice, respectively. Embryos were imaged using a light microscope with an attached Retiga 2000R monochrome camera (QImaging: Surry, BC, Canada) with an attached X-Cite 120Q to detect fluorescence.

Crown–rump length (CRL) was recorded for each specimen using ImageJ [23]. CRL is defined in this study as the maximum cranial-caudal dimension of an embryo, which typically runs from the most superior/posterior portion of the head to the base of the tail in the sagittal plane (Figure 1). CRL measurements were acquired for each embryo twice on different days and had to meet an intra-observer measurement error standard of no greater than 0.1 mm to be included in the study. The two CRL measurements were then averaged to establish a single CRL score for each embryo.

2D outlines of the hindlimb of each embryo were used to estimate limb staging ages. Briefly, a photograph of the hindlimb was uploaded to the embryonic Mouse Ontogenetic Staging System (eMOSS) [3]. The 2D outline of the limb of each embryo is traced and then compared with a library of mean hindlimb shapes for each hour of development between E10–E15 to provide an estimate of an embryo’s limbstaging age [3]. Whenever possible, both the left and right hindlimbs were staged and their average was used to estimate limbstaging age. All statistical analyses for this work were performed using R [24]. Determination of the average time of appearance for each ossification center in the Osx-GFP and C57BL/calcein samples were estimated using the R package OptimalCutpoints [25]. OptimalCutpoints uses receiver operating characteristic (ROC) curves to establish boundaries between binary variables (presence/absence of ossification centers) based on their relationship to another variable (limbstaging age or CRL) and accuracy measures selected by the researcher (sensitivity/specificity ratio).

The use of two different samples of mice to compare the level of bone development with crown–rump length and limbstaging age provides two advantages. First, osteoblast differentiation and bone mineralization are distinct developmental processes with osteoblast differentiation preceding mineralization, and it is possible that each process has a unique individual relationship with size and shape. Second, the two samples cover two embryonic time periods, which allows examination of the relationship between size and shape and of the progress of bone development over a larger proportion of developmental time.

## Figures and Tables

**Figure 1 jdb-06-00031-f001:**
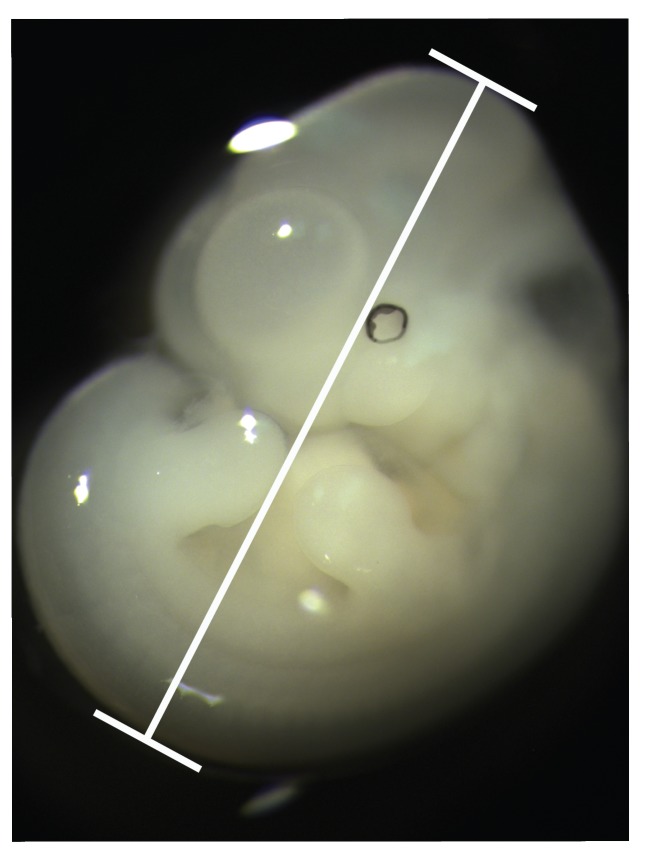
Crown-rump length measures the maximum total length of an embryo using a sagittal view.

**Figure 2 jdb-06-00031-f002:**
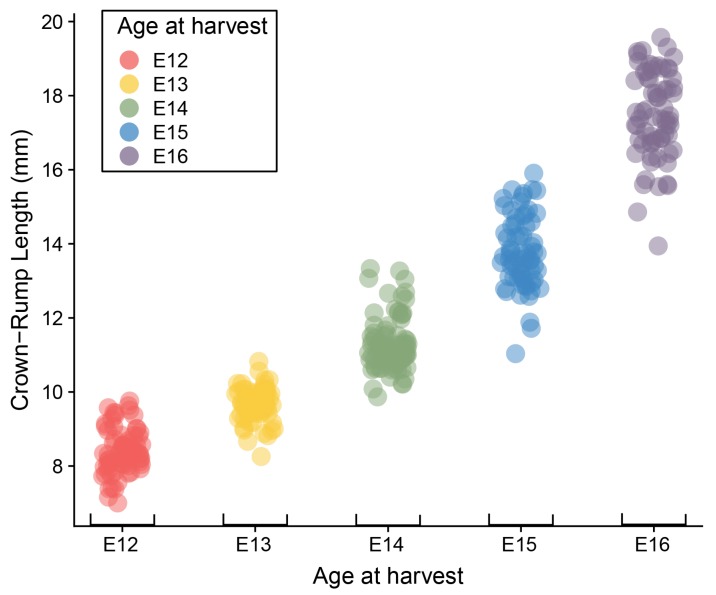
Plot of Crown-rump length by age at harvest. Dots represent individual specimens and are colored according to age at harvest.

**Figure 3 jdb-06-00031-f003:**
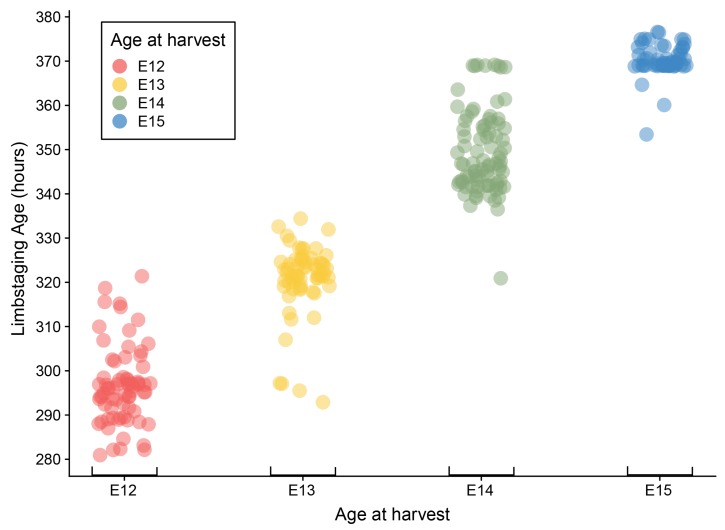
Plot of limbstaging age by age at harvest. Dots represent individual specimens and are colored according to age at harvest.

**Figure 4 jdb-06-00031-f004:**
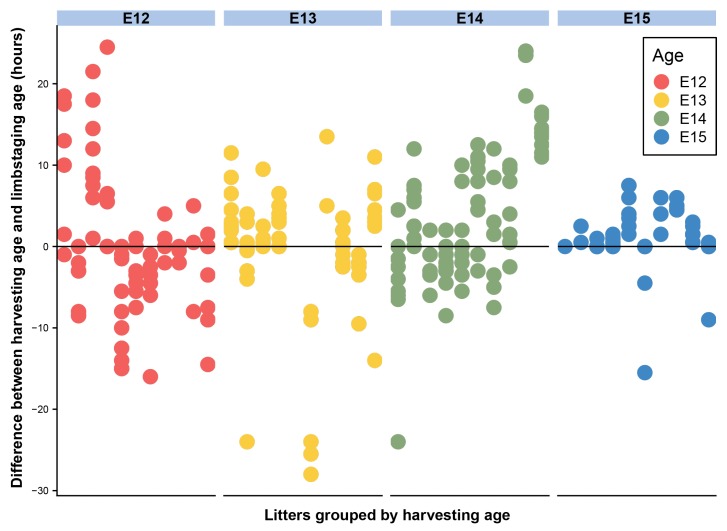
Deviation between age at harvest and limbstaging age for E12–E15. Each column represents a litter, with each data point representing an individual embryo. Colors represent age at harvest. Scores on the *y*-axis represent the difference between the expected age of an embryo based on age at harvest and limbstaging age for each individual.

**Figure 5 jdb-06-00031-f005:**
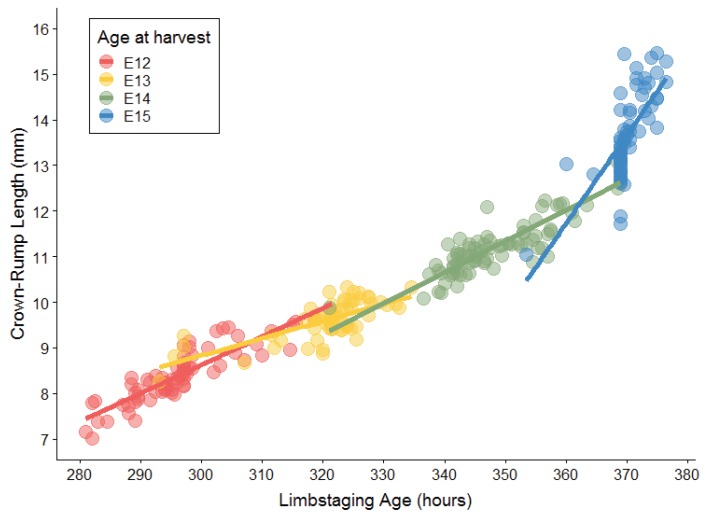
Graph of crown–rump length and limbstaging age. Dots represent individual specimens and are colored according to age at harvest.

**Figure 6 jdb-06-00031-f006:**
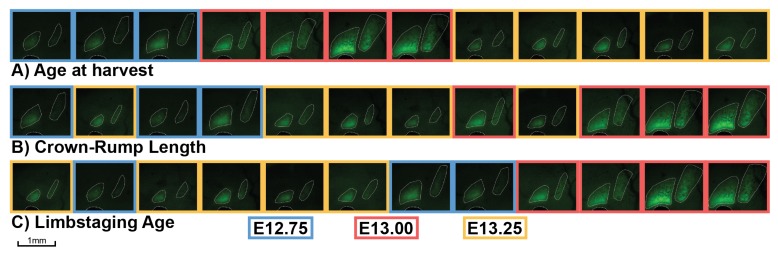
Appearance of frontal and parietal bones in Osx-GFP mice ordered by age at harvest, CRL, and limbstaging age. Each image shows a lateral view of the cranial vault of a left-facing Osx-GFP embryo, with Osx expression revealed by green fluorescent protein. The frontal and parietal bones and the eye are outlined in white to show the extent of Osx expression in each bone. The boxes outlining each specimen are colored according to age at harvest. Each age group is represented by a single litter. (**A**) specimens ordered according to age at harvest; (**B**) specimens ordered according to CRL; (**C**) specimens ordered according to limbstaging age.

**Figure 7 jdb-06-00031-f007:**
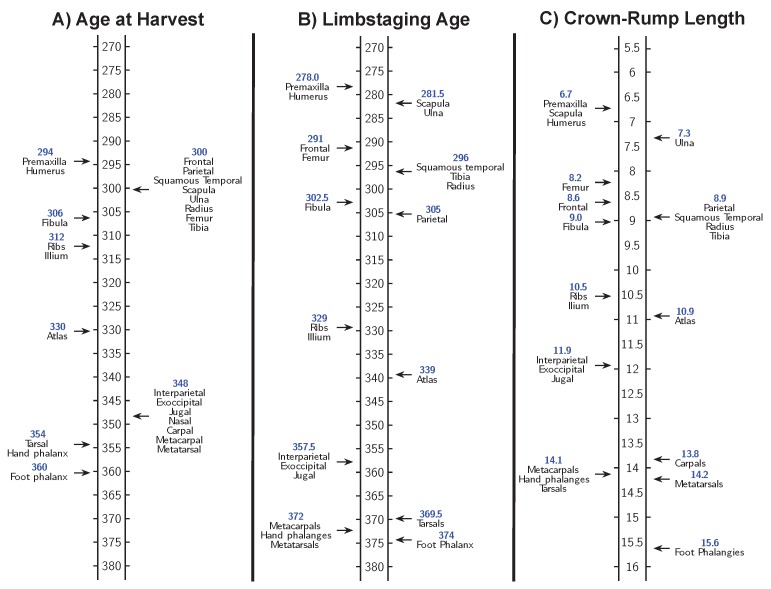
Timeline of ossification center appearance marked by osteoblast differentiation (expression of Osx) generated using Osx-GFP mice ordered according to: (**A**) age at harvest; (**B**) limbstaging age; and (**C**) CRL.

**Figure 8 jdb-06-00031-f008:**
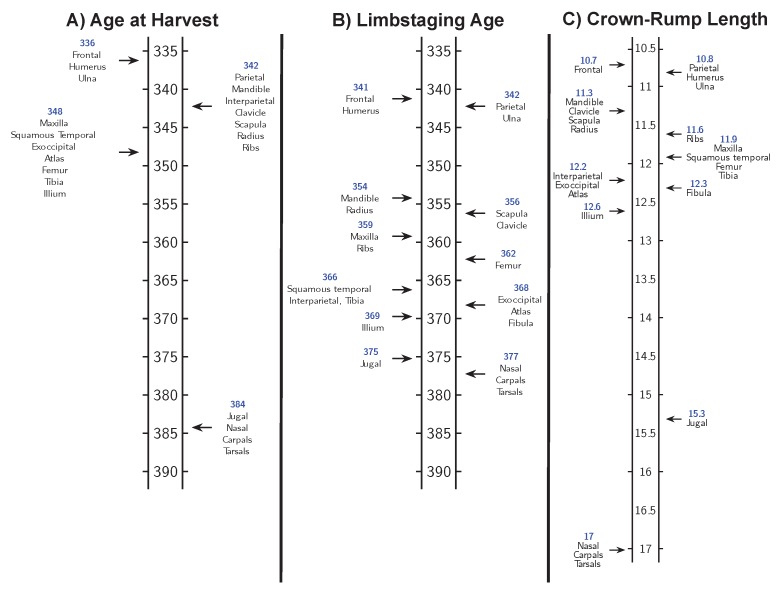
Timeline of ossification center appearance marked by mineralization of osteoid (marked by calcein staining) generated using C57BL/6-calcein mice ordered according to: (**A**) age at harvest; (**B**) limbstaging age; and (**C**) CRL.

**Table 1 jdb-06-00031-t001:** Summary statistics for crown–rump length.

Harvesting Age	Mean	SD	Min	Max	Range
E12	8.41	0.61	7.00	9.75	2.75
E13	9.61	0.43	8.26	10.33	2.07
E14	11.29	0.71	9.87	13.33	3.46
E15	13.70	0.94	11.03	15.46	4.43
E16	17.49	1.19	13.94	19.57	5.63

**Table 2 jdb-06-00031-t002:** ANOVA for Crown-Rump Length. Statistically significant *p*-values **bolded**.

	Df	Sum Sq	Mean Sq	F Value	Pr (>F)
Limbstaging Age	1	1062.36	1062.36	10261.27	**0.0000**
Harvesting Age	1	1.21	1.21	11.71	**0.0007**
Litter	38	70.83	1.86	18.00	**0.0000**
Uterus Location	1	0.02	0.02	0.22	0.6375
Residuals	238	24.64	0.10		

**Table 3 jdb-06-00031-t003:** ANOVA for limbstaging Age. Statistically significant *p*-values **bolded**.

	Df	Sum Sq	Mean Sq	F Value	Pr (>F)
Limbstaging Age	1	8703.86	8703.86	612.08	**0.0000**
CRL	1	214,058.47	214,058.47	15,053.19	**0.0000**
Litter	38	7342.67	193.23	13.59	**0.0000**
Uterus Location	1	54.67	54.67	3.84	0.0511
Residuals	238	3384.39	14.22		

**Table 4 jdb-06-00031-t004:** Pearson’s correlation values for Crown-Rump Length and Developmental Age.

Harvesting Age	Correlation	R^2^
E12	0.89	0.79
E13	0.73	0.53
E14	0.90	0.80
E15	0.84	0.71
Overall	0.96	0.92

**Table 5 jdb-06-00031-t005:** Sample sizes for each embryological age. Each embryo was collected at 9:00 a.m. on the designated embryological day.

Age	E12	E13	E14	E15	E16
**N=**	71	65	86	67	67

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
