# Peer review of "It’s about Time: Ossification Center Formation in C57BL/6 Mice from E12–E16"

_jdb, 2018, doi:10.3390/jdb6040031_

Round 1

Reviewer 1 Report

This is an interesting manuscript that Flaherty & Richtsmeier have presented to establish CRL and eMOSS to determine skeletal development in Embryonic C57BL/6 mice. Staging embryonic mice timelines is a critical factor in understanding developmental cues not only in skeletal tissue but in other tissue and organ development.

The introduction, methods and results are clearly and concisely presented. The discussion could be improved to compare their new methods of assessing embryo development compared to standard research published protocols. 

Shape & size have been used as the standards to measure the developmental parameters in this submission. Have the authors looked at the weight of the embryo (or is there any literature evidence) in regards to the development time of the ossification centre forming and hence altering the developmental age of the embryo?

In Fig7 & 8 it is not clear as to why 2 methods were used to compare ossification centre and the age at harvest / Limbstaging Age / crown-rump length. Are the authors trying to make a comparison between early marker expression of the ossification centre and late mineralization marker of the ossification centre?

Scale bars should be added to the images of Fig 6

Author Response

We were unable to collect weight data for the embryos in this study because it would have either required us to weight them wet (either from amniotic fluid or PBS), which, given the size of the embryos used this study, would have introduced too much error to be considered reliable data, or weigh them dry, which would have desiccated the embryos and introduced the risk of altering the shape of the hindlimb as a result of the drying processes. It would be possible to collect weight data in the course of another research project that does not require the collection of hindlimb outline data.

Other researchers rarely report the use of staging methodology in their methods, so it is difficult to find directly comparable uses of staging methodology to contrast with our own. However, we have added a discussion of our methods compared to other staging methodologies to the Discussion section to provide more context for our readers.

For figures 7 and 8, we were providing tables for these two aspects of ossification center appearance (osteoblast differentiation and bone mineralization) both because we believed that the tables would be individually useful, and because it would be useful to assemble data regarding the delay between the onsets of osteoblast differentiation and bone mineralization.

We have placed a scale bar for the images of figure 6 as requested.

Reviewer 2 Report

I enjoyed reading the research article about the “It’s about time: Ossification center formation in C57BL/6 mice from E12-E16”. In this study, Flaherty and Richtsmeier tried to develop a method to trace the temporal sequences of mice development. Which is vital for research pertaining to growth and development.

The authors used two methods; change in size (using crown-rump length) and change in shape (using eMOSS limbstaging ages) to establish a detailed sequence of cranial ossification center appearance in C57BL/6 and in Osx-GFP mice. The Osx (Osterix) gene expression was used to established sequences of ossification center appearance. Then they compared the ossification center appearance in each method. However, the time lines were different in these two methods.

The authors concluded that each developmental events have its own unique timeline and it is important carefully consider the general developmental principals and particular organ or system development when designing the research.

In general, this study provides important insights to use of murine models in developmental biology research. I have one comment to improve the quality of the discussion.

1.    Since there are many previous literatures on staging system of mouse development. The authors can discuss some of them in their discussion and can compare and contrast them in more detail with their current method.

Author Response

We have added a discussion of our methods compared to other staging methodologies to the Discussion section. Other researchers rarely report the use of staging methodology in their methods, so it is difficult to find directly comparable uses of staging methodology to contrast with our own, but we have discussed it such as we are able.

Reviewer 3 Report

Flaherty et al. showed the variability of growth and development in mouse embryos harvested according to ages. They examined two parameters, size and shape, by investigating crown-rump length (CRL) and embryonic Mouse Ontogenic Staging System (eMOSS), respectively; the latter revealed limbstaging ages. They showed that even at the same age, sizes and limbstaging ages showed variations, suggesting that the age is not reliable parameter to consider the developmental stage. It is notable that sizes are more variable in later ages, whereas limbstaging ages are less; limbstaging ages are more variable in earlier stages. They also demonstrated that the CRL and limbstaging ages provided more reasonable sequence of ossification center expansion than age at harvest. Overall, the topic is interesting in this field, and the manuscript is well written.

Author Response

Thank you for reviewing our manuscript and for your kind appraisal of it.